# Spatio-Temporal Description of the NDVI (MODIS) of the Ecuadorian Tussock Grasses and Its Link with the Hydrometeorological Variables and Global Climatic Indices

**Jhon Villarreal-Veloz** [1,2], **Xavier Zapata-Ríos** [1,2,*] [ID], **Karla Uvidia-Zambrano** [1] **and Carla Borja-Escobar** [1]

1    Department of Civil and Environmental Engineering, Escuela Politécnica Nacional,
     Ladrón de Guevara E11-253, Quito P.O. Box 17-01-2759, Ecuador; jhon.villarreal@epn.edu.ec (J.V.-V.)
2    Center for Research and Water Resources Studies (CIERHI), Escuela Politécnica Nacional,
     Ladrón de Guevara E11-253, Quito P.O. Box 17-01-2759, Ecuador
*    Correspondence: xavier.zapata@epn.edu.ec

**Abstract:** This study examined the changes in tussock grass greenness over 18 years (2001–2018) using NDVI data from 10 key areas of the Páramo ecosystem in the Ecuadorian Andes. In addition, the study investigated the influence of hydrometeorological variables (precipitation, soil temperature, and water availability) and climatic indices (AAO, MEI, MJO, NAO, PDO, El Niño 1 + 2, 3, 3.4, and 4) on greenness dynamics. The spatial and temporal variations of NDVI were studied, applying several analysis and indicators, such as: the standard deviation, *z*-score anomalies, Sen slope, Mann–Kendall test, and time integrated-NDVI (TI-NDVI). Linear and multilinear correlations were used to evaluate the influence of hydrometeorological variables and climatic indices on the greenness of tussock. The findings of the study show that Páramo, located in the Inter-Andean valley above 2° S, is the most productive, followed by those located in the Royal Range (eastern cordillera). The anomalies and trends of NDVI on the Royal Range tended to be greening over time. NDVI showed a moderate multilinear correlation with precipitation and soil temperature, and a strong response to water availability. Finally, NDVI was weakly linearly related to the climatic indices, the most representative being the MJO, and slightly related to ENSO events. Understanding the regional and global-scale variables that control tussock grasses' phenology will help to determine how present and future climate changes will impact this ecosystem.

**Keywords:** NDVI; tussock grasses; grasslands; paramo; precipitation; global climate indices

## 1. Introduction

Along the Andes Mountain range in its main north–south direction, gradients in climate are observed because this mountain chain is exposed to different patterns of atmospheric circulation and various global climatic phenomena. Consequently, the western and eastern slopes of the Andes are characterized by contrasting climatic asymmetries [1]. This climatological variability in the region is manifested by variability in the seasonal and spatial patterns of precipitation and air temperature [2]. In the equatorial Andes, moisture sources originate from the Amazon Basin and the tropical Atlantic from the east, and from the Pacific Ocean from the west [3,4]. In this section of the Andes, precipitation amounts are greatest north of the equator, on eastern slopes, and at low to medium elevations [4]. In the inter-Andean valleys, located between the western and eastern cordillera (Royal Range), two rainy seasons are common from February to May and October to November as a consequence of the two moisture sources, together with the Inter-Tropical Convergence Zone oscillation [5]. Interannual and interdecadal climate variability is also observed because of large-scale phenomena such as the El Niño Southern Oscillation (ENSO) [2].

The knowledge of the climate in the Ecuadorian Andes is largely based on meteorological observations derived from field stations [2,4,6]. However, a limited number of stations

with a short observation period are located at elevations above 3000 m.a.s.l. The National Institute of Meteorology and Hydrology of Ecuador (INAMHI) shows in its database a total of 34 meteorological stations (13.6% of all stations in the country) with elevations above 3000 m.a.s.l. However, only 50% of these stations are currently in operation. On the other hand, information derived from regional and global climate models has allowed an understanding of the climate at regional scales in the Andes [7]. Unfortunately, these models still report climate variability in the Andes with high uncertainty due to their coarse spatial resolution and inability to capture the strong climate gradients and topographical heterogeneity in the region [8,9]. Consequently, there are still relevant processes in mountain areas that are not fully understood. For example, there are interactions between climate and vegetation [9]. This knowledge gap includes how changes in climate variability (e.g., ENSO) influence ecohydrological processes and the spatial extent of ENSO influence in the Andes [4,10,11]. Regrettably, mountain ecosystems such as the high Andes are expected to be some of the most affected by climate change [12]. Consequently, there is a need to investigate these issues and fill existing knowledge gaps.

The seasonal variation of vegetation observed by remote sensing has been considered as a simple proxy for many ecosystem processes that take place on the planet [13–15]. The structures and processes that occur in terrestrial ecosystems are highly dependent on climate [16]. Specifically, the variability of the spectral indices of the land surface vegetation has been very useful in understanding aboveground net primary production [17–19], phenology [14], canopy structure [20], agricultural processes and silvicultural [21], soil carbon stocks [17,22,23], water availability [17], droughts [17,24], energy cycles [25,26], and biodiversity [27–29], among others. A specific application has been the understanding of the relationship between vegetation responses to climate and large circulation patterns [30–33]. Therefore, the seasonal variation in vegetation over the land surface observed by remote sensing is considered a good indicator of environmental change [34] and a broad indicator of climate variability [32,35]. Temporal changes within an ecosystem can be monitored by the greenness variability referred to in this study as the vegetation dynamics [36]. As a result, satellite information has been an important tool for understanding vegetation dynamics at regional and global scales [37–39].

A large number of satellite sensors have been used to obtain vegetation indices. The Moderate Resolution Imaging Spectroradiometer (MODIS) is one of the most common sources of information. MODIS has provided data for ecosystem monitoring with appropriate spatial and temporal resolutions and improved geometric and radiometric properties [40,41]. The spatial greenness dynamics were derived using MODIS Normalized Difference Vegetation Index (NDVI) time series data [32]. This vegetation index is the most widely used indicator for monitoring vegetation from satellites [42]. Positive and negative NDVI trends have been found in various vegetation classes in southern South America, including the Andes. The Multivariate El Niño Index of the Southern Oscillation (ENSO) and the Index of Antarctic Oscillation (AAO) explain the variability in annual phenology in some regions of southern South America [32]. In tropical South America, it was observed that the NDVI responds slightly to ENSO events [43], as does the NDVI over the Colombian Andes [44]. In Ecuador, Silvia [45] studied the temporality of NDVI and concluded that the vigor of the plant is greater in the Amazon region, followed by the Andean and coastal regions. In addition, NDVI studies on wetlands in the southern coastal region of Ecuador have demonstrated a direct relationship between NDVI and biomass with precipitation and air temperature. In contrast, in a small central Andean sector of Ecuador, satellite information (NDVI, CWSI: Crop Water Stress Index and SAVI: Soil Adapted Vegetation Index) made it possible to determine the best form of irrigation (drip irrigation) for *Opuntia ficus* [46].

The tropical Andes are a global conservation hotspot because of their large number of endemic plants and the degree of threat to their ecosystems [47]. Páramos belong to one of these ecosystems and extend from 2800 to 4700 m.a.s.l [48]. These extend between latitudes 11° N and 8° S over 35,000 km$^2$ [49]. This ecosystem is essential because it provides

important environmental services, such as water regulation and supply, carbon storage, soil formation, biodiversity conservation, climate change mitigation, recreational services, and food, among others [1,50]. In addition, it houses endemic animals, a diversity of flora, and indigenous cultures that contribute to planetary ecosystem diversity [51]. Rapid population growth, deforestation, mining and subsistence farming, drainage of wetlands, frequent burning of vegetation, and changes in land cover are threatening the high Andean ecosystems in the Andes [52,53].

Existing studies on the Páramo throughout the Ecuadorian Andes lack a detailed analysis of the response of vegetation to different global climatic factors. A quantitative evaluation of the changes in vegetation has the potential to define the interaction between climate and the response of Páramo vegetation. In this study, the Ecuadorian Andes were defined at 1° N and 5° S latitudes. This section of the Andes is characterized by rapid changes in elevation over short distances. NDVI was utilized, which provides continuous vegetation monitoring with spatial and temporal consistency, allowing a comparison of Vegetation Index (VI) dynamics between different study areas within a region of interest. Frequent and regular updates of the vegetation index and its high spatial and temporal resolutions are among the advantages of using VIs over other hydrometeorological variables.

Throughout the world, strong relationships have been found between vegetation indices and climatic variables such as precipitation and air temperature [54,55]. Satellite-derived precipitation and air temperature have also been available since the late 1970s. However, these estimates are still inaccurate for mountainous topographies [56,57]. Nevertheless, this relationship is dependent on geographic location; therefore, analyses of different ecosystems should be performed. It is widely recognized that meteorological variables such as precipitation and air temperature have a significant influence on the spectral indices of vegetation [58]. However, few studies have examined these patterns in the Ecuadorian Andes.

This study postulates that the dynamics of vegetation in the Páramo ecosystem, located at high elevations in the Andes responds to climatic conditions and water availability. In addition, global weather patterns such as El Niño Oscillation should influence the greenness dynamics of this ecosystem. This postulate is based on the case of tropical glaciers which have been widely studied as indicators of climate variability in mountain ecosystems [59,60], and the widely recognized impact of ENSO on the dynamics of tropical glaciers [61]. Therefore, the main objective of this research is to characterize the seasonal patterns and the interannual variability of the dynamics of vegetation greenness in the Páramo ecosystem using MODIS-NDVI in a period of time between the years 2001 to 2018. In the context of this general objective, the following research questions were formulated: (i) What is the spatial and temporal variability of NDVI at different locations within tussock grasses throughout the equatorial Andes? (ii) How do the vegetation indices respond to changes in rainfall, soil temperature and water availability? (iii) Is there any temporal trend in the NDVI time series over the 18 years of observations? (iv) How does NDVI respond to different global climate indices like MEI, AAO and El Niño indices?

## 2. Materials and Methods

### 2.1. Study Area

Ecuador is located in the north-western corner of South America (Figure 1a), and is crossed by the Andes Mountain range, which runs from north to south. The Andes in Ecuador are made up of two main mountain ranges: The Royal Range or Eastern Range and the Western Range. Both mountain ranges hydrologically divide Ecuador into two main basins: the Pacific Ocean and Amazon/Atlantic. In addition, the Andes divide the territory of Ecuador into three regions (Coastal, Andean and Amazon), which are characterized by their own seasonality (Table 1). The mountain range extends to 6° latitude (1.2° N–4.8° S), with an approximate area of 60,400 km$^2$, which corresponds to 24% of the surface of Ecuador. The elevation of this region is in a range from 1500 to 6310 m.a.s.l. [62]. In the Andean region

there is a population of approximately 6.1 million inhabitants with an economy based on various activities: (i) agriculture, where the main products are potatoes, strawberries, wheat, barley and carrots; (ii) cattle raising, reaching 48.4% of Ecuador's production [63]; and (iii) mining, where the main extracted resources are gold, silver, bronze, antimony, lead, zinc and platinum [64]. Additionally, the Andes have an ideal environment for the diversification of multiple ecosystems because of their geomorphological characteristics, soil types, precipitation patterns, and temperature gradients [65]. Within this territory, 6 main ecosystems have been identified: Páramo, Humid Puna, Dry Puna, Cloud Forest, Seasonal Andean Forest, and Dry Andean Forest [66].

The Páramos develop in humid and sub-humid regions and in climatic conditions characterized by low or freezing temperatures and high radiation because of their location at high elevations [1,48]. Although the climate in this ecosystem is fairly uniform, extreme daily changes are common [48]. The moisture sources originate from the Amazon region and/or the warm and cold ocean currents of the Pacific Ocean [3,4,48]. Precipitation patterns are generally bimodal with a central dry season and have a typical range of 800–2000 mm/year [6,48,67]. Andosols with a deep A horizon and high concentrations of organic matter are typically found in Páramos [68,69]. The soils are dark in color, acidic, have a high cation exchange capacity, and are generally poorly developed [67].

Focusing on the vegetation of the Páramo ecosystem, Harling [70] defined three types of Páramo in Ecuador: (i) tussock, (ii) cushions, and (iii) desert Páramos. The tussock grasses are between 3000 and 4000 m.a.s.l (Figure 2) and are dominated by *Calamagrostis*, *Festuca* and *Stipa*, with the presence of meadows and pads. The cushion Páramos occupy areas above 4000 m.a.s.l. where the grasslands are replaced by *Azorella pedunculata*, meadows, thickets, and rosettes represented by *Espeletia pycnophylla*. Desert or sandy Páramos that present restrictive conditions for vegetation growth are located at elevations greater than 4300 m.a.s.l. In the present study, tussock grass Páramos have been selected as an ideal sentinel ecosystem for the analysis of the dynamics of the vegetation and its relationship with the climate, showing its predominance in the Ecuadorian Andes, due to the fact that they extend from north to south in an approximate area of 9650 km$^2$, equivalent to 64% of the surface of the Ecuadorian Páramo.

**Table 1.** Rainy and dry seasons for the three regions of continental Ecuador [71].

|  | Coast | Highlands | Amazon |
| --- | --- | --- | --- |
| Average annual air temperature (°C) | 25.5 | 12.7 | 21.8 |
| Average annual rainfall (mm/year) | 892.9 | 798.9 | 3449.4 |
| Rainy season | January to April | Mar to April and October to November | Almost the whole year [1] |
| Dry season | June to December | May to September and December to January | August to January |

[1] Heavy rains occur from May to August.

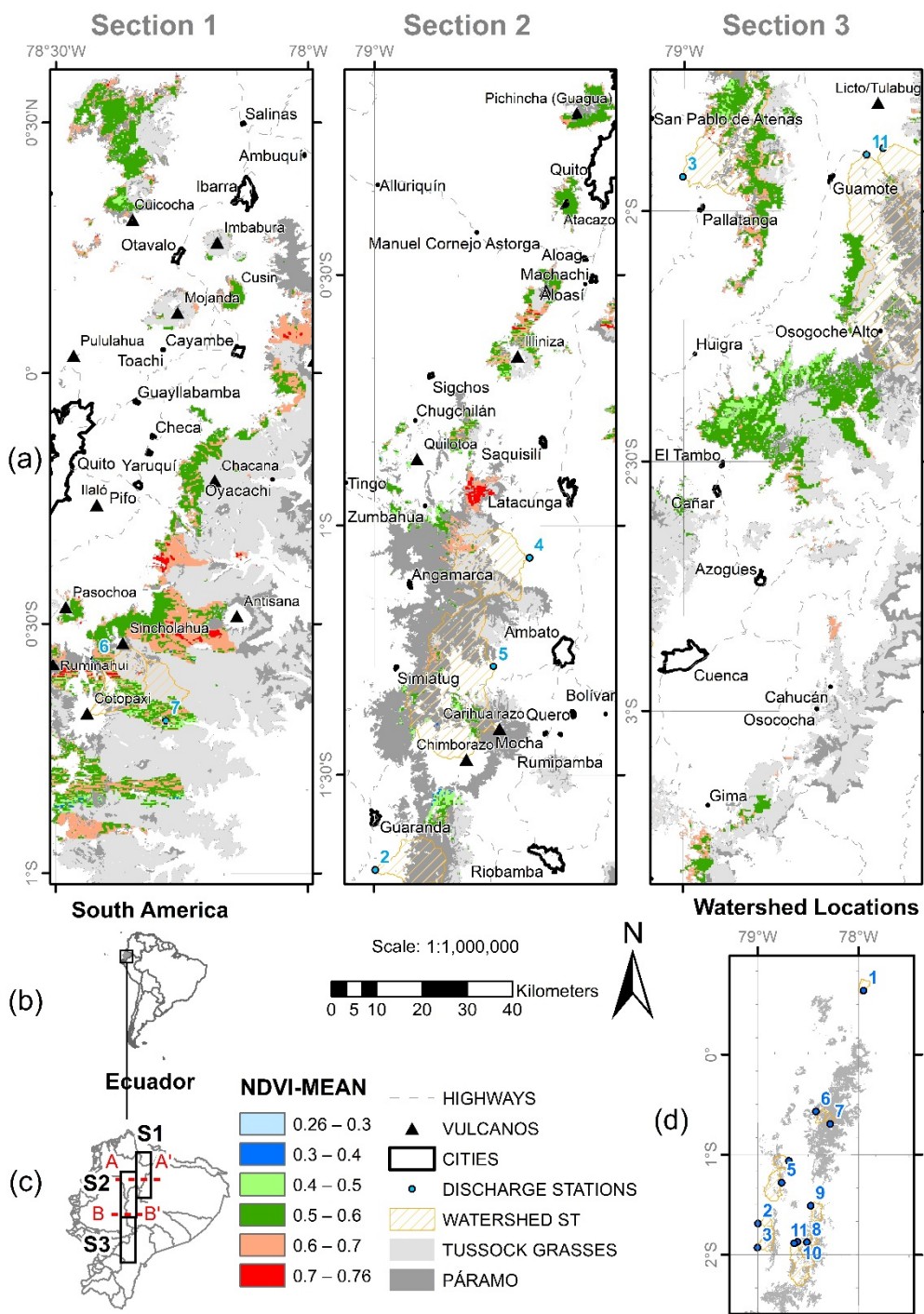

**Figure 1.** Location of the Páramos in Ecuador. (**a**) Three panels for easy visualization of the study area. These were numbered from north to south. The locations of the hydrological stations (turquoise points) and their watersheds (orange polygons) are displayed, along with the average NDVI (Normalized Difference Vegetation Index) for the period 2001–2018. (**b**) Location of Ecuador in South America. (**c**) Location of the Tussock Páramo ecosystem in, the cuts A-A' and B-B', and the location of the ecuadorian hydrological basins. S1–S3 sections indicated in (**a**). (**d**) Location of the 12 high-elevation hydrological basins that have more than 50% of their surface covered by the tussock Páramo ecosystem.

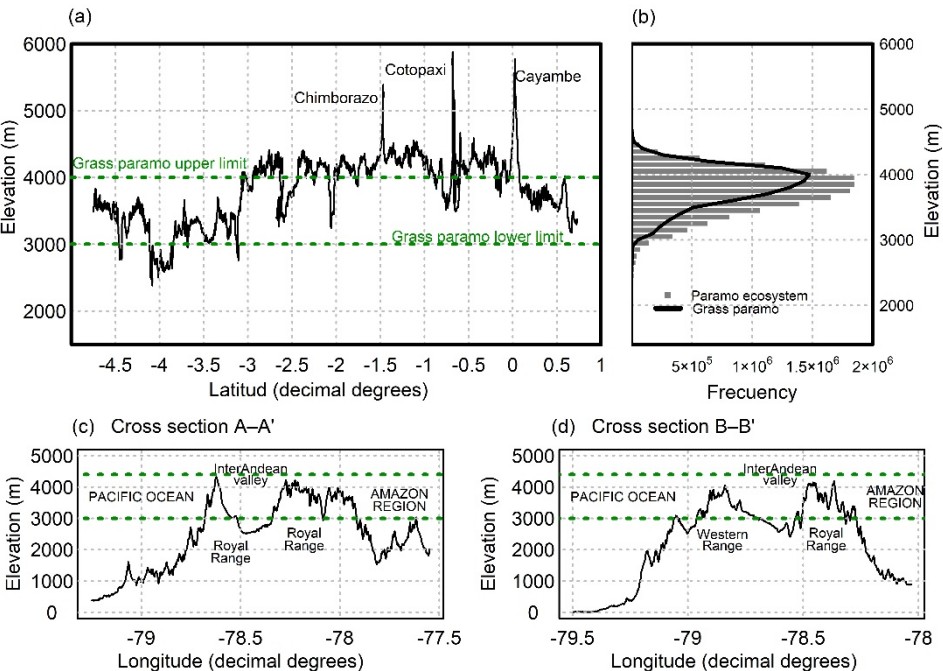

**Figure 2.** (**a**) Elevation profile along the continental divide. (**b**) Histogram of elevations derived from 30 m DEM for both the Páramo ecosystem and the tussock Páramo. (**c**) Elevation profile of cross section A–A's where the altitudinal extension of the tussock Páramo ecosystem can be observed in both the western and eastern mountain ranges. (**d**) Elevation profile of cross section B–B's and altitudinal extension of the tussock Páramo ecosystem in the western and eastern mountain ranges. The location of the cross sections A–A' and B–B' can be seen in Figure 1. Green lines shown the upper and lower elevation limits of the paramo ecosystem.

*2.2. Time Series of Vegetation Indices, Climate Information and Global Teleconnection Indices*

2.2.1. NDVI Dataset

NDVI is an indicator of vegetation response that has been used in multiple ecosystem studies [44,72–75]. NDVI is sensitive to chlorophyll in leaves and is calculated in terms of reflectance at red and near-infrared wavelengths, according to Equation (1) [13].

$$\text{NDVI} = \frac{\rho_{NIR} - \rho_{red}}{\rho_{NIR} + \rho_{red}} \tag{1}$$

where $\rho_{NIR}$ and $\rho_{red}$ are the bidirectional surface reflectances within the MODIS bands [13]. An interpretation of the type of vegetation cover from a range of NDVI values is shown in Table 2.

**Table 2.** NDVI range values and general interpretation according to vegetation density [76].

| NDVI Value | Cover Type |
| --- | --- |
| −1.0–0.1 | Sterile area |
| 0.1–0.5 | Vegetative cover |
| 0.5–0.7 | Dense vegetation |
| 0.7–1.0 | Very dense vegetation |

The NDVI information for the Ecuadorian Páramos was obtained using the MODIS Collection 6 product MOD13Q1 available between 2001 and 2018. MOD13Q1 is a spatial, spectral and radiometric representation of surface vegetation conditions [77]. NDVI products were generated from MODIS data using the 16-day peak composite method and combined with the MODIS bidirectional reflectance distribution function to obtain

a higher percentage of clear-sky data that minimizes noise. The MODIS images have a spatial resolution of 0.002° (~250 m) and were downloaded from the Distributed Active Archive Center (DAAC) Level-1 and Atmosphere Archive & Distribution System (LAADS; https://ladsweb.modaps.eosdis.nasa.gov/ (accessed on 4 March 2019)). Subsequently, these images were projected onto WGS84 using ArcGIS 10.4. In this ArcGIS analysis, it was possible to generate 414 mosaics and create a database for the Ecuadorian Páramos (h9v9, h10v8, h10v9) based on the Páramos shown on the land use coverage map at the national scale, developed by the Ministry of the Environment of Ecuador [78].

Consecutively, the Páramo mosaics were analyzed with the R-cran 3.5.3 software and the images were filtered with the reliability layer of the MOD13Q1 product to maintain only cells with good quality and eliminate negative values. Thus, an annual NDVI time series was created with a total of 414 values per cell corresponding to the 18 years of study as a result of the integration of 23 images per year (1 image every 16 days). Missing information was spatially and temporally filled in by pixels that showed at least 50% of the total temporal information. Spatial approximation was carried out using the Ordinary Kriging method, which generates a reliable performance in the prediction of missing information on the images [79]. To fill in the information with this method, the experimental variogram was calculated, a theoretical variogram was selected between Gaussian, exponential, pentaspherical or wave for each image, and the NDVI values were predicted on the missing spatial information. Finally, the temporal approximation was performed through the Savitzky—Golay filter within the TIMESAT 3.3 software [30,80].

### 2.2.2. Satellite Climate Information and Water Availability

The University of California, Santa Barbara—Climate Hazards Group InfraRed Precipitation with Station (UCSB-CHIRPS) project provides precipitation data with a spatial resolution of 0.01° (~1 km). This information is based on satellite infrared rainfall and rain gauge station observations [81]. Monthly precipitation data for the study area was downloaded from http://chg.geog.ucsb.edu/data/chirps/ (accessed on 25 Febrary 2019). These images were then stitched, cropped, and resampled (bilinear interpolation) at a spatial resolution of 250 m to match the extent and spatial resolution of NDVI. Precipitation images were then converted to ASCII files to create a data matrix.

Land Surface Temperature and Emissivity (LST&E) were derived from the MODIS-MOD11B3 product with a spatial resolution of 0.055° (~6 km). The information in this product is based on the split window method, where the result is an accurate soil surface temperature [82]. The monthly temperature information was downloaded from the portal https://lpdaac.usgs.gov/products/mod11b3v006/ (accessed on 16 August 2019), it was transformed into Celsius degrees and processed in the same way as precipitation.

Water discharge rates at specific points in the Andes were used as indicators of water availability in the region. Information on water availability for 12 high-altitude basins was provided by the INAMHI (Table 3). The Páramo ecosystem is the dominant land cover type within the selected hydrographic basins, with coverage exceeding 50% of its surface (Figure 1).

### 2.2.3. Global Teleconnection Indices

Planetary-scale waves are the product of atmospheric processes regulated by mountain ranges and the boundaries between land and water masses. These global phenomena generate anomalies in the climatic seasons and, in long-term periods, influence large geographic regions [83]. Several global climatic anomalies were selected, including the Antarctic Oscillation (AAO), ENSO Multivariate Index (MEI), and Madden–Julian Oscillation (MJO). Table 4 shows a summary of the teleconnection rates studied and the digital repositories where they were downloaded.

**Table 3.** INAMHI surface water discharge stations at high elevations. The Páramo ecosystem is the dominant land cover type covering more than 50% of the surface within the hydrographic basins.

| Code | Name | Lat (°) | Long (°) | Elevation (m) | Area (km$^2$) | % of Páramo | Data |
|------|------|---------|----------|---------------|----------------|-------------|------|
| H0064 | El Ángel en Puente Ayora | 0.6375 | −77.9518 | 2889 | 124.77 | 56.0 | 2003–2013 |
| H0333 | San Lorenzo en San Lorenzo | −1.6901 | −78.9978 | 2438 | 106.80 | 66.9 | 2000–2013 |
| H0337 | Pangor Aj Salto | −1.9319 | −79.0028 | 1480 | 280.05 | 50.1 | 2000–2013 |
| H0793 | Cusubamba | −1.0644 | −78.6922 | 2962 | 181.12 | 65.9 | 2000–2013 |
| H1143 | Ambato en Mazanahuaico | −1.2824 | −78.7636 | 3018 | 450.67 | 57.5 | 2005–2013 |
| H0158 | Pita Aj Salto | −0.5710 | −78.4240 | 3550 | 127.32 | 80.8 | 2000–2009 |
| H0722 | Yanahurco Dj Valle | −0.6953 | −78.2825 | 3606 | 87.06 | 98.3 | 2000–2013 |
| H0787 | Alao en Hda. Alao | −1.8772 | −78.5117 | 3200 | 114.60 | 72.3 | 2000–2013 |
| H0788 | Puela Aj. Chambo | −1.5122 | −78.4747 | 2475 | 208.03 | 52.2 | 2000–2013 |
| H0789 | Guargualla Aj. Cebadas | −1.8739 | −78.6052 | 2828 | 189.38 | 73.0 | 2004–2013 |
| H0790 | Cebadas Aj. Guamote | −1.8872 | −78.6384 | 2840 | 707.38 | 62.3 | 2000–2013 |
| H0896 | Matadero en Sayausi | −2.8766 | −79.0730 | 2602 | 299.51 | 82.5 | 2000–2013 |

**Table 4.** Description of the studied teleconnection indices.

| Name | Years/ Resolution | Definition/Website |
|------|-------------------|--------------------|
| Antarctic Oscillation (AAO) | 1979–present Monthly | Empirical orthogonal function to the 1000-hPa mean height anomaly. www.cpc.ncep.noaa.gov/products/precip/CWlink/daily_ao_index/aao/ (accessed on 21 February 2020) |
| ENSO Multivariate Index (MEI) | 1950–present Monthly | Principal component of sea pressure level, zonal and meridional components of surface wind, sea surface temperature, surface air temperature, and cloud cover. psl.noaa.gov/enso/mei/ (accessed on 21 February 2020) |
| Madden–Julian Oscillation (MJO) | 1978–present Daily | A pair of empirical orthogonal functions of the combined fields of averaged 850-hPa zonal wind, 200-hPa zonal wind, and satellite-observed outgoing longwave radiation. www.cpc.ncep.noaa.gov/products/precip/CWlink/daily_mjo_index/proj_norm_order.ascii (accessed on 30 January 2020) |
| North Atlantic Oscillation (NAO) | 1950–present Monthly | Rotated principal component analysis on 500 mb height anomalies. www.cpc.ncep.noaa.gov/products/precip/CWlink/pna/nao.shtml (accessed on 23 January 2020) |
| Pacific Decadal Oscillation (PDO) | 1854–present Monthly | Spatial average of the monthly sea surface temperature in the Pacific Ocean north of 20° N. www.ncdc.noaa.gov/teleconnections/pdo/ (accessed on 24 January 2020) |
| Niño 1 + 2 | 1948–present Monthly | Sea surface temperature in the El Niño 1 + 2 region. psl.noaa.gov/data/correlation/nina1.data (accessed on 22 January 2020) |
| Niño 3 | 1948–present Monthly | Sea surface temperature in the El Niño 3 region. psl.noaa.gov/data/correlation/nina3.data (accessed on 22 January 2020) |
| Niño 4 | 1948–present Monthly | Sea surface temperature in the El Niño 4 region. psl.noaa.gov/data/correlation/nina4.data (accessed on 22 January 2020) |
| Niño 3.4 | 1948–present Monthly | Sea surface temperature in the El Niño 3.4 region. psl.noaa.gov/data/correlation/nina34.data (accessed on 22 January 2020) |

*2.3. Methods*

A NDVI spatio-temporal analysis was performed and summarized within 10 defined zones within the study area. The zones were divided with respect to their predominant terrain orientation and location, resulting in the salient climatic influence of the Pacific Ocean, Amazon, or a hybrid regime (Figure 3). These zones are defined by the summit lines

of the western and eastern mountain ranges as well as by the inter-Andean alley delimited by both mountain ranges. In addition, a differentiation is made between the northern and southern parts of the Páramo, defined by latitude 2° S.

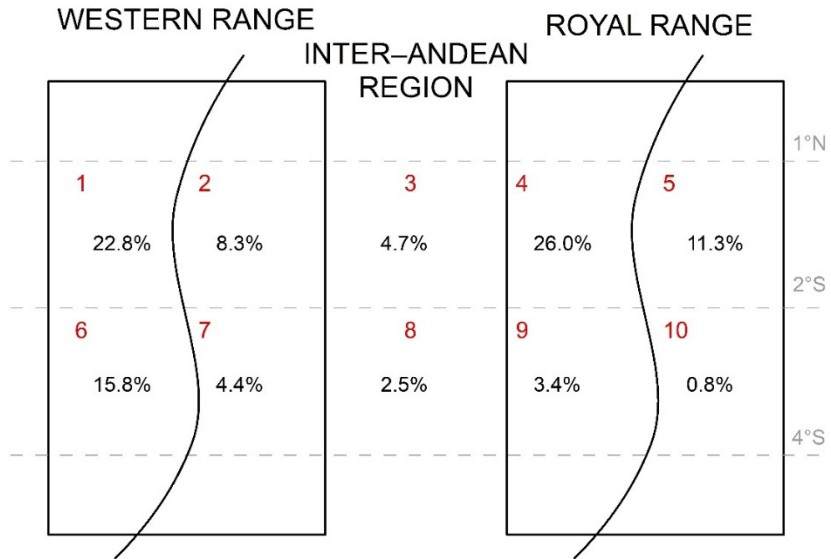

**Figure 3.** Schematization of the spatial division of the study area with its corresponding percentage of the total area. The central line of the western and eastern mountain range represents their summit line. Zone 1: West of the Northern Western Range; zone 2: East of the Northern Western Range; zone 3: Northern Inter-Andean Valley; zone 4: West of the Northern Royal Range; zone 5: East of the Northern Royal Range; zone 6: West of the Southern Western Range; zone 7: East of the Southern Western Range; zone 8: Southern Inter-Andean Valley; zone 9: West of the Southern Royal Range; zone 10: East of the Southern Royal Range.

### 2.3.1. Spatio-Temporal Analysis of NDVI

A productivity indicator called Time Integration—NDVI (TI-NDVI) was applied in this study. TI-NDVI is defined as the area under the curve of the NDVI time series [84]. The TI-NDVI was calculated for each cell to describe the spatial distribution of the productivity of the tussock grasses. In addition, the variability of vegetation dynamics per cell was calculated using the standard deviation (Equation (2)). The NDVI anomalies were computed using the *z*-score (Equation (3)) to illustrate the spatial and temporal distributions. Additionally, the monthly anomalies were smoothed with a moving average of a 36-month average to partially remove the inter-annual variability of the tussock grasses. The moving average was calculated using Equation (4), where i corresponds to the month within the study period. The Mann–Kendall test [85,86] and its slope with 95% confidence were calculated on the NDVI time series to determine whether there was a spatial trend for each cell. The magnitude of the trend was estimated using Sen's slope where a positive slope value indicates a monotonic increase or an increasing trend, while a negative value of the slope indicates a monotonic decrease or a decreasing trend [87]. The maximum value of TI-NDVI during a quarter analysis was developed to determine the period (December–January–Febrary: DJF; March–April–May: MAM; June–July–August: JJA; September–October–November: SON) where the maximum NDVI value generally occurs. This last analysis requires comparing the sum of the TI-NDVI within each quarter to select the period of greatest occurrence that presents the maximum annual productivity.

$$SD(x,y) = \sqrt{\frac{\sum (NDVI(x,y,t) - MEAN(NDVI))}{N-1}} \tag{2}$$

$$Z\ SCORE(x,y,t) = \frac{NDVI(x,y,t) - MEAN(NDVI)}{SD(NDVI)} \qquad (3)$$

$$MEAN(Z\ SCORE)_i = \frac{\sum_i^{i+36} Z\ SCORE_i}{36} \qquad (4)$$

where *NDVI(x,y,t)* is the NDVI value at a given position and time. *MEAN*(*NDVI*) is the mean NDVI value of the database. *N* is the number of months during the study period. *SD*(*NDVI*) is the standard deviation of the NDVI database. *Z SCORE(x,y,t)* is the time anomaly of each cell.

### 2.3.2. NDVI Analysis—Climatic Variables and Water Availability

Precipitation and soil temperature are the main variables controlling phenology and ecosystem production [12]. A Pearson correlation coefficient with 5% significance was calculated to understand the relationship between vegetation and climatic variables [88]. A simple and multivariate correlation was performed between NDVI and precipitation or soil temperature, and both climatic variables. The correlation was calculated at different time levels: monthly, bi-monthly, quarterly, four-monthly, half-yearly and annually. Similarly, water availability is directly related to vegetative growth to a greater or lesser extent. Therefore, a simple linear regression model was used to determine the relationship between the mean monthly discharge and mean monthly NDVI time series.

### 2.3.3. NDVI Analysis—Global Climate Indices

The Global Teleconnection Indices (GTI) allowed statistical analysis between them and the NDVI time series. Table 4 shows the GTIs that were selected to understand their influence on the grassland Páramos. Initially, the Pearson correlation with 5% significance between NDVI and GTI was calculated to later calculate the cross-correlation ($p$ value < 0.05) with a delay of one period to the past of the GTI for the monthly, semi-annual and annual series.

## 3. Results

Tussock grasses are located along 9650 km$^2$ in the western and royal mountain range between 3000 and 4000 m.a.s.l. (Figure 2). Sixty-eight percent of this vegetation drains towards the Amazon basin and the remaining 32% flows into the Pacific Ocean. A total of 1242 MODIS images were downloaded and processed from 2001 to 2018. After filtering the MODIS images and filling in the data gaps, only a quarter of the surface extent of the tussock Páramo ecosystem resulted in good quality NDVI data, corresponding to a total of 39,192 cells. Seventy-three percent (73.1%) of the total observations were in the northern part (north of 2° S latitude) and 26.9% in the southern part. Approximately fifty-one percent (51.3%) of all observations are in the Western Range and 41.5% in the Royal Range.

### 3.1. Spatio-Temporal Analysis of NDVI

Table 5 shows the mean elevation, mean annual precipitation, mean monthly ground surface temperature, mean values, median, standard deviation, and temporal trend analysis of the NDVI calculated and summarized for the defined zones within the study area (Figure 3). The average greenness level of the grasslands (Figure 1) shows that the grassland Páramo ecosystem of the Western Range and the northern Inter-Andean Valley are slightly greener than their corresponding ones to the south. On the other hand, the eastern mountain range shows the same greenness both to the north and to the south (4, 5, 9 and 10 zones).

**Table 5.** Elevation, precipitation and land surface temperature summarized for the defined zones within the study area.

| No. | Study Area % | Elevation | | Precipitation | | Land Surface Temperature | |
|---|---|---|---|---|---|---|---|
| | | Mean | SD | Mean | SD | Mean | SD |
| 1 | 22.8 | 291.9 | 291.9 | 778.5 | 107.2 | 17.6 | 2.7 |
| 2 | 8.3 | 202.1 | 202.1 | 619.8 | 82.2 | 17.9 | 3.0 |
| 3 | 4.7 | 190.9 | 190.9 | 911.0 | 123.6 | 19.5 | 3.1 |
| 4 | 26.0 | 208.9 | 208.9 | 946.9 | 134.2 | 16.3 | 3.5 |
| 5 | 11.3 | 245.3 | 245.3 | 1285.1 | 186.4 | 15.7 | 4.0 |
| 6 | 15.8 | 267.9 | 267.9 | 436.1 | 64.0 | 18.3 | 3.5 |
| 7 | 4.4 | 211.4 | 211.4 | 577.2 | 76.7 | 17.4 | 3.7 |
| 8 | 2.5 | 230.0 | 230.0 | 799.5 | 108.4 | 17.5 | 3.2 |
| 9 | 3.4 | 232.5 | 232.5 | 933.8 | 131.0 | 18.3 | 3.1 |
| 10 | 0.8 | 131.6 | 131.6 | 944.6 | 132.5 | 19.4 | 3.0 |

The average zone variability of the tussock grasses greenness indicated by means of standard deviation can be seen in Table 6—Column 4. This shows little variability in greenness in general, where the grasslands located to the north are more variable than those in the south, with higher values in the Amazon basin. On the other hand, the southern grasslands showed the greatest variability in the western mountain range.

**Table 6.** Average of the mean, median, standard deviation, *z*-score, Mann–Kendall, Sen Slope and TI-NDVI in each of the areas shown in Figure 3. Trimestral Picks: March-April-May (MAM), June-July-August (JJA).

| No. | Mean | Median | SD | z-Score | Mann Kendall | Sen Slope | TI-NDVI | Trimestral Picks |
|---|---|---|---|---|---|---|---|---|
| 1 | 0.56 | 0.57 | 0.071 | −0.17 | 0.12 | 0.00016 | 3715.8 | MAM |
| 2 | 0.58 | 0.58 | 0.074 | 0.01 | 0.13 | 0.00016 | 3802.1 | MAM |
| 3 | 0.60 | 0.60 | 0.071 | 0.26 | 0.09 | 0.00012 | 3926.0 | MAM |
| 4 | 0.59 | 0.59 | 0.070 | 0.22 | 0.13 | 0.00016 | 3906.7 | MAM |
| 5 | 0.59 | 0.60 | 0.078 | 0.24 | 0.15 | 0.00018 | 3915.7 | MAM |
| 6 | 0.54 | 0.54 | 0.069 | −0.43 | 0.16 | 0.00018 | 3585.6 | MAM |
| 7 | 0.55 | 0.55 | 0.055 | −0.37 | 0.15 | 0.00014 | 3610.7 | MAM |
| 8 | 0.57 | 0.57 | 0.053 | −0.05 | 0.08 | 0.00009 | 3767.2 | JJA |
| 9 | 0.59 | 0.59 | 0.052 | 0.13 | 0.21 | 0.00020 | 3857.3 | MAM |
| 10 | 0.59 | 0.59 | 0.047 | 0.14 | 0.25 | 0.00025 | 3860.1 | MAM |

The mean zone *z*-score anomalies of the tussock grasses are shown in Table 6—Column 5. The negative (positive) anomalies represent areas of lower (higher) vegetation with respect to the mean of tussock grasses. Positive and negative anomalies are distributed in space by 46.5% and 56.5%, respectively. In the northern part of the grasslands, there are positive anomalies with the exception of the western mountain range (1 and 2 zones), which presents a negative and neutral anomaly for the western and eastern parts, respectively. In contrast, the southern tussock grasses in the western mountain-range (6 and 7 zones) have the largest negative anomalies, whereas the western part of the western mountain range being the dominant anomaly. On the contrary, the northern and southeastern mountain ranges (4, 5, 9 and 10 zones) present positive anomalies, with the northern part showing the highest productivity.

The monthly NDVI anomalies of the study area and the 36-month moving average are shown in Figure 4. The moving average shows the tendency of grasslands to be greener over time, with maximum anomalies of up to 0.15 SD. However, during the period 2009–2011, the grasslands experienced a decrease in vegetation greenness, values close to 0. Similarly, tussock grasses showed higher (lower) greenness in the face of La Niña (El Niño) events.

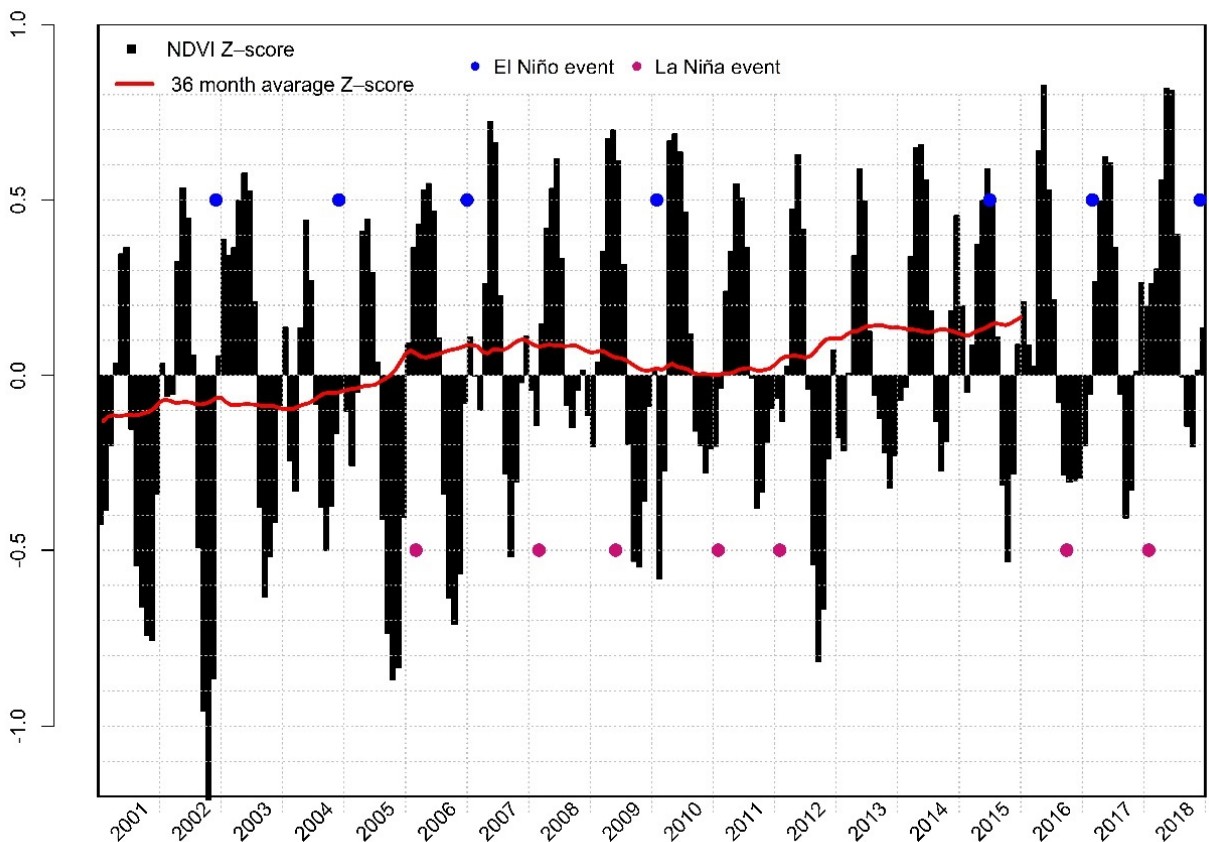

**Figure 4.** Comparison of the monthly mean of the NDVI anomalies of the study area (black bars) and the 36–month moving mean of the NDVI anomalies (red line) of the study area averaged for the period 2001–2018. El Niño and La Niña events are represented by blue and fuchsia dots, respectively.

Table 6—Column 6 shows the mean Mann–Kendall trend test results for each zone with a significance level of 5% for the period 2001–2018. The Tau coefficient of the trend test ranges from −1 to 1 and is read as Spearman's coefficient [89]. The trend results were discretized as described by Yan [90]. The trend test shows that 56.26% of the study area shows statistically significant trend results. For the entire study area, a trend of 5.75% negative and 50.51% positive was observed. Within all zones, there is a very weak positive trend. Of these, the greatest trends are found in the eastern mountain range, both in the northern and southern tussock grasses.

Figure 5 shows the quarter where the maximum NDVI productivity values of the tussock grasses usually occur. According to the summary of results presented in Table 6, it is observed that in general the grasslands present their maximum annual quarterly productivities in the MAM period, and only the southern inter-Andean Valley (zone 7) presents it in JJA.

### 3.2. NDVI Analysis—Climatic Variables and Water Availability

The CHIRPS product shows a distribution of 147–2214 mm/year of precipitation over the grassland Páramo ecosystem. The highest precipitation occurs over the Amazon basin around Antisana volcano (section 1) and towards the southwest of La Merced de Buenos Aires city at the northern end of the Western Range. The standard deviation of the precipitation is low over the entire study area. However, the highest SD values for precipitation are observed around Guagua Pichincha volcano. On the other hand, the soil temperature has a range of 6.2–25.0 °C where the highest values are shown in the lower limit of the grassland and the lowest are found around the peak flow line. In addition, the smallest standard deviations of soil temperature are found in the Pacific basin and the largest in the Amazon basin.

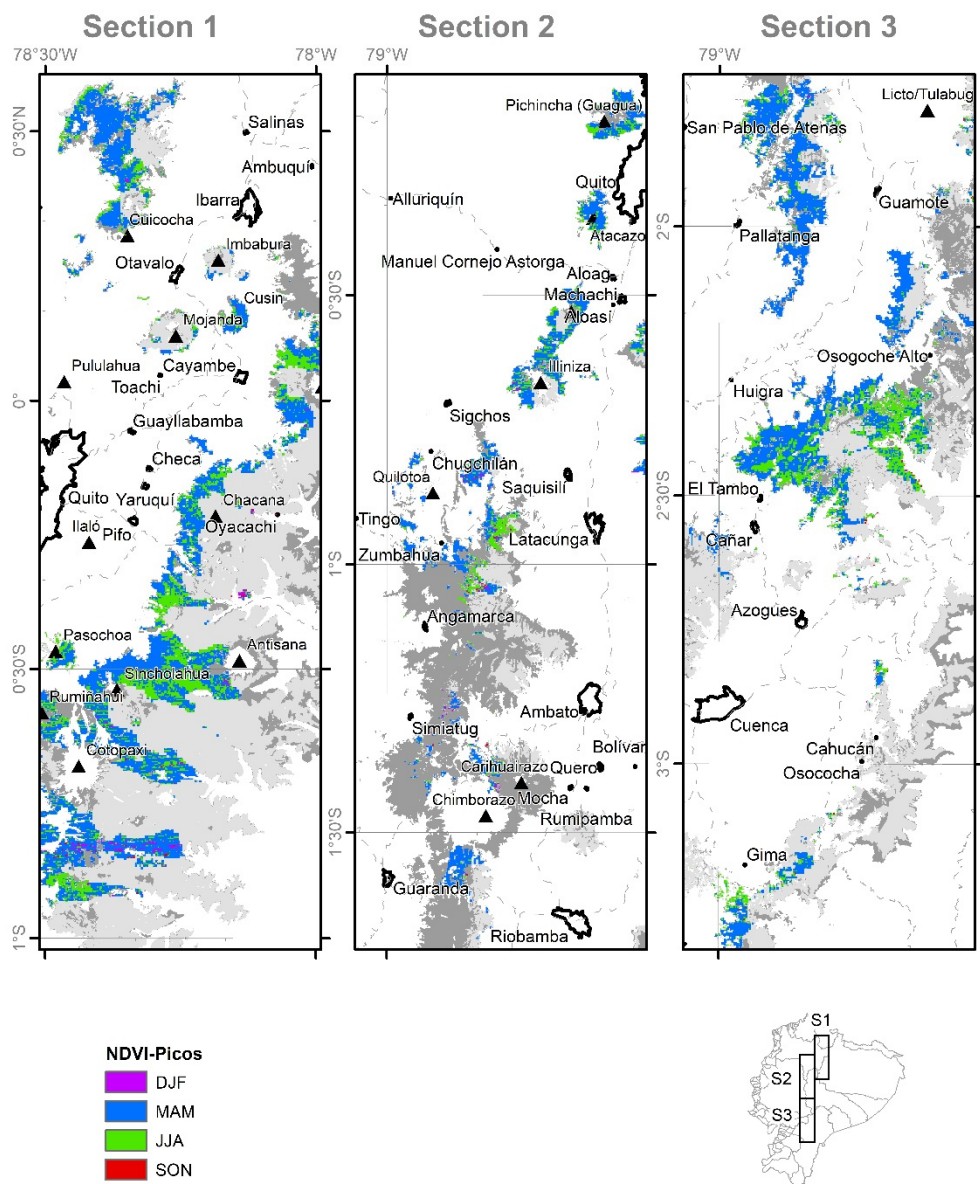

**Figure 5.** Peaks of quarterly maximums of NDVI in the grasslands. For each pixel, the quarter where the maximum NDVI values usually occur has been selected. The year was divided into the following quarters: DFJ, MAM, JJA, and SON according to Ecuadorian seasonality. Black Boxes S1–S3 indicated the section number.

Table 7 shows the spatial average of the simple and multivariable correlations between NDVI and precipitation, temperature and both variables. It can also be seen in Table 7 the amount of significant data out of the total for each level of correlation (monthly, bimonthly, quarterly, four-monthly, half-yearly and annual). The six-month correlations show the strongest Pearson correlations in combination with a high percentage of significant data. In such a way, the six-monthly correlations have been averaged for each zone and are shown in Table 8.

**Table 7.** Summary of correlation between NDVI and climatic variables. Pearson's correlation was applied at various temporal levels (monthly, bimonthly, quarterly, four-monthly, half-yearly and annually). Only are presented correlations with a significant level of 95%.

| Temporary Scale | Precipitation | | Temperature | | Prec. + Temp. | |
|---|---|---|---|---|---|---|
| | Mean Corr. | % Data | Mean Corr. | % Data | Mean Corr. | % Data |
| 1 month | 0.147 | 44 | −0.414 | 50 | 0.428 | 53 |
| 2 months | 0.308 | 58 | −0.429 | 47 | 0.495 | 52 |
| 3 months | 0.409 | 73 | −0.446 | 40 | 0.527 | 50 |
| 4 months | 0.451 | 55 | −0.190 | 13 | 0.567 | 51 |
| 6 months | 0.480 | 48 | −0.544 | 36 | 0.620 | 45 |
| 12 months | 0.280 | 8 | −0.540 | 9 | 0.630 | 26 |

**Table 8.** Average of the six-monthly Pearson correlation of NDVI—Precipitation, NDVI—Temperature and NDVI—Precipitation + Temperature of the significant cells corresponding to each zone shown in Figure 3. For each zone, the percentage of significant correlations with respect to the cells within them.

| No. | NDVI—Precipitation | | NDVI—Temperature | | NDVI—Prec. + Temp. | |
|---|---|---|---|---|---|---|
| | Mean Corr. | % Data | Mean Corr. | % Data | Mean Corr. | % Data |
| 1 | 0.52 | 73 | −0.57 | 57 | 0.68 | 66 |
| 2 | 0.47 | 50 | −0.50 | 28 | 0.60 | 43 |
| 3 | 0.47 | 35 | −0.48 | 24 | 0.57 | 47 |
| 4 | 0.40 | 25 | −0.51 | 27 | 0.57 | 35 |
| 5 | 0.41 | 32 | −0.45 | 19 | 0.49 | 32 |
| 6 | 0.52 | 70 | −0.58 | 62 | 0.66 | 66 |
| 7 | 0.44 | 50 | −0.45 | 15 | 0.53 | 22 |
| 8 | 0.41 | 34 | 0.00 | 0 | 0.44 | 3 |
| 9 | 0.42 | 45 | −0.45 | 2 | 0.55 | 3 |
| 10 | 0.41 | 32 | −0.42 | 7 | 0.68 | 10 |

Table 8—Column 6 summarizes the Pearson coefficients of the multivariate correlations between NDVI, precipitation, and temperature for the 10 zones. The correlations are weak (11.79%), moderate (17.68%), strong (15.06%) and very strong (0.70%). The grasslands, with the exception of the west of the western mountain range present a moderate positive correlation (weak positive correlation with precipitation). Negative correlations were obtained between NDVI and temperature. Above latitude 2° S, in zones 1,2, 4, there are moderate negative correlations, unlike the weak negative correlation of the inter-Andean alley and the east of eastern mountain range. The multivariate correlation of NDVI with precipitation and temperature shows moderate positive correlations, and in the case of the west of the western mountain range, close to a strong correlation. On the other hand, the number of significant correlations of the southern eastern mountain range in terms of temperature is negligible, so the results do not represent these areas.

There are clear differences in the water discharge in the hydrographic basins that drain the western and eastern mountain ranges (Figure 6). For example, basins located in the western mountain range showed increases in discharge around April and May and decreases around September and October. Some basins that drained the Western Range even exhibited a bimodal pattern (for example, H0064, H0793, H0333, H0337, and H0896). While the basins located in the eastern mountain range were characterized by a maximum value in the middle of the year (for example, H0722, H0788, H0787, H0789, and H0790). It was also evident that the water discharge in the Royal Range is greater than that in basins that drain the Western Range. The peak discharge and base flow in the basins draining the Royal Range doubled the peak discharge and base flow in the basins draining the Western Range (Figure 6).

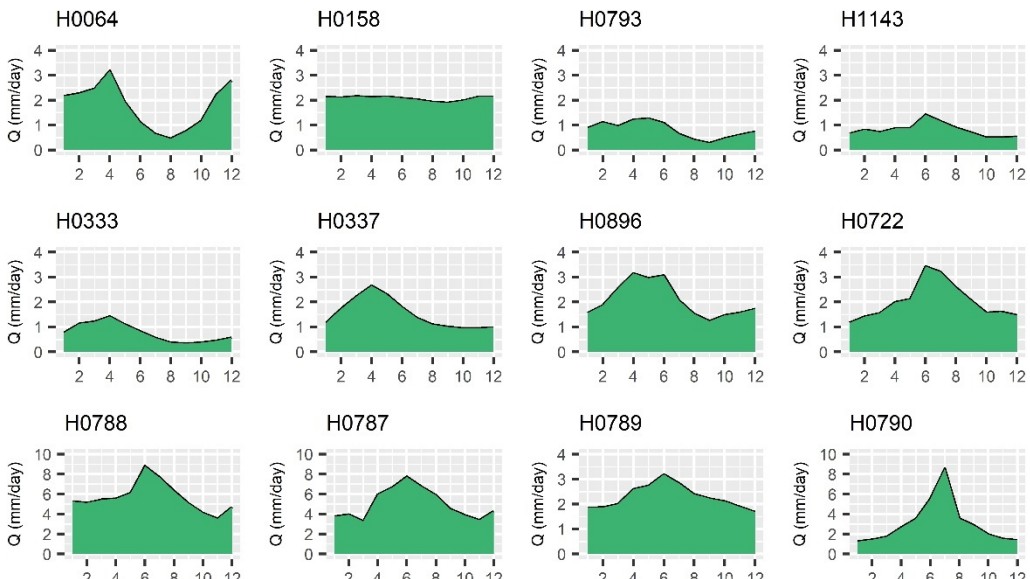

**Figure 6.** Monthly discharge registered in 12 high-elevation hydrographic basins that have more than 50% of their surface covered by the Páramo ecosystem. The download was provided by the National Institute of Meteorology and Hydrology (INAMHI).

The temporal variability of NDVI is closely associated with water availability, as indicated by the surface water discharge. Table 9 shows the Pearson correlation coefficients ($R^2$) resulting from a linear regression model between the mean monthly surface water discharge and the mean monthly variability of the NDVI. There was a moderately explanation for the relationship between the discharge recorded in the western watersheds and the mean NDVI time series. The relationship was the weakest between the surface water discharge recorded in the eastern basins and the mean NDVI time series. One of the reasons is that the data set for both indices has more data available along the Western Range than along the Royal Range. Therefore, western VI values predominate in both datasets. Later, the mean monthly surface water discharge was correlated with the mean NDVI time series for each group indicated by the blue line in Figure 1.

**Table 9.** Pearson's correlation coefficients ($R^2$) as a result of a linear regression between mean monthly discharge flows and the total mean of NDVI. Linear regression was performed between the total mean of NDVI, the mean of the VI time series for each group.

| | Mean NDVI | |
|---|---|---|
| **Stations** | **Western Range** | **Royal Range** |
| | **$R^2$** | **$R^2$** |
| H0064 | 0.06 | - |
| H0333 | 0.45 * | - |
| H0337 | 0.55 * | - |
| H0896 | - | 0.50 * |
| H0793 | - | 0.43 * |
| H1143 | - | 0.20 |

*p* value: * < 0.05;

### 3.3. NDVI Analysis—Global Climate Indices

In the same way as the analysis between NDVI and climatic variables, the Global Teleconnection Indices have been analyzed at different levels of temporal information (monthly, bimonthly, quarterly, four-monthly, half-yearly, and annual). From this analysis, the results at the semester level present the highest correlations in combination with the amount of significant information. The semi-annual Pearson correlation between the NDVI

and GTI is summarized in Table 10, where the AAO, MJO and NAO indices have the lowest percentages of significant data. Of these indices, MJO presents higher average correlation coefficients than the other indices. On the other hand, the cross-correlation analysis of a GTI lag for the monthly, semi-annual and annual series has lower Pearson coefficients than the analysis without time shift.

**Table 10.** Average of the spatial correlation between NDVI and the Global Teleconnection Indices (GTI). Pearson's correlation with a significance level of 95% was applied in each cell.

| No. | AAO | | MEI | | MJO | | NAO | | PDO | |
|---|---|---|---|---|---|---|---|---|---|---|
| | Mean R | % Data | Mean R | % Data | Mean R | % Data | Mean R | % Data | Mean R | % Data |
| 1 | 0.23 | 76% | 0.19 | 26% | 0.20 | 31% | 0.32 | 91% | 0.20 | 69% |
| 2 | 0.23 | 74% | 0.19 | 20% | 0.22 | 38% | 0.32 | 82% | 0.25 | 69% |
| 3 | 0.24 | 67% | 0.19 | 24% | 0.23 | 39% | 0.30 | 86% | 0.27 | 62% |
| 4 | 0.24 | 73% | 0.19 | 21% | 0.21 | 33% | 0.29 | 74% | 0.23 | 50% |
| 5 | 0.25 | 74% | 0.19 | 25% | 0.26 | 25% | 0.31 | 72% | 0.21 | 41% |
| 6 | 0.21 | 68% | 0.26 | 67% | 0.18 | 10% | 0.38 | 96% | 0.33 | 91% |
| 7 | 0.21 | 37% | 0.23 | 69% | 0.20 | 13% | 0.35 | 87% | 0.29 | 73% |
| 8 | 0.24 | 26% | 0.27 | 67% | 0.19 | 20% | 0.30 | 77% | 0.24 | 51% |
| 9 | 0.26 | 66% | 0.19 | 39% | 0.18 | 17% | 0.31 | 87% | 0.27 | 76% |
| 10 | 0.25 | 68% | 0.19 | 33% | 0.14 | 2% | 0.30 | 91% | 0.28 | 90% |

| No. | EL NIÑO 1 + 2 | | EL NIÑO 3 | | EL NIÑO 4 | | EL NIÑO 3.4 | |
|---|---|---|---|---|---|---|---|---|
| | Mean R | % Data | Mean R | % Data | Mean R | % Data | Mean R | % Data |
| 1 | 0.21 | 55% | 0.20 | 54% | 0.20 | 20% | 0.19 | 34% |
| 2 | 0.20 | 49% | 0.19 | 41% | 0.20 | 18% | 0.18 | 25% |
| 3 | 0.21 | 45% | 0.20 | 42% | 0.20 | 25% | 0.19 | 30% |
| 4 | 0.23 | 57% | 0.20 | 49% | 0.20 | 18% | 0.19 | 30% |
| 5 | 0.25 | 65% | 0.22 | 51% | 0.20 | 21% | 0.20 | 33% |
| 6 | 0.25 | 76% | 0.27 | 85% | 0.26 | 58% | 0.25 | 74% |
| 7 | 0.22 | 65% | 0.23 | 75% | 0.25 | 68% | 0.23 | 70% |
| 8 | 0.24 | 59% | 0.29 | 71% | 0.28 | 68% | 0.29 | 70% |
| 9 | 0.23 | 66% | 0.21 | 58% | 0.20 | 38% | 0.20 | 44% |
| 10 | 0.24 | 84% | 0.20 | 55% | 0.19 | 28% | 0.18 | 32% |

## 4. Discussion

### 4.1. Spatio-Temporal Analysis of NDVI

The analysis shows that tussock grasses have a similar average behavior throughout the study area with grasslands above 2° S latitude showing higher productivity, higher precipitation, and lower surface temperature. The mean, median, z–score and TI–NDVI indicate that the northern inter-Andean alley (zone 3) is the most productive zone, followed by the Royal Range (Table 6). Zone 3 high productivity is a consequence of grassland response to rainfall, water availability, and soil temperature (Table 5). Another variable that can contribute to productivity is elevation as has been indicated by [91,92].

The seasonality of the average productivity of the NDVI generally presents its maximum peak in the period March–April–May, corresponding to the first and most intense cycle of precipitation in the region [43,93]. On the other hand, the maximum annual productivity of the Southern Inter-Andean Valley (zone 8) occurs in June–July–August, corresponding to the summer (winter) of the Ecuadorian highlands (Amazon). Zone 8 can also be affected by the humidity of the Amazon, which presents an increase in its precipitation, which in turn is influenced by the Atlantic Ocean [4]. In other words, the grasslands of zone 8 have a relationship with the Amazon and the surface temperature of the Atlantic Ocean [4,94].

The NDVI variability (Table 6) shows that zones above 2° S latitude are more variable than their southern counterparts. Of all the zones, zone 5 has the highest variability (0.078) in the study area, which also corresponds to the highest variability of precipitation

(186.3 mm/year) and soil temperature (4.0 °C) in the study area (Table 5). These results are in agreement with those of studies on the Royal Range in Colombia, which show the greatest variation in NDVI is linked to variations in precipitation and air temperature. On the other hand, the tussock grasses of zones 7 to 10 show less variability, indicating the influence of moisture transported from the Amazon [95]. Unlike zone 6, which has a similar variability with its counterpart to the north (zone 1) and shows no influence of humidity from the Amazon due to the blockade of the Western Range [96].

The spatial and temporal anomalies of NDVI (Table 6) shown that the greatest anomalies occur in the tussock grasses of the northern areas and of these, the anomalies of the Western Range are positive and for the Royal Range, positive. In addition, these results are in agreement with those obtained in the Mann–Kendall and Sen's Slope analysis, which show the Royal Range with the largest positive trends. That is, the anomalies and trends show that within the 2001–2018 period, there is an increase in the greenness of the tussock grasses on the Royal Range due to a possible effect of $CO_2$ fertilization [97,98]. On the other hand, it is observed that zones 3 and 8 have the lowest greening tendencies (zone 3: $\tau = 0.09$; zone 8: $\tau = 0.08$) together with the highest positive anomaly and a slight negative anomaly (zone 3: $z$-score = 0.26; zone 8: $z$-score = $-0.05$), respectively. This observation can be interpreted as the maximum extent of productivity in these areas and the small positive trends may be the response to global $CO_2$ fertilization [98]. Finally, in a peculiar way, it can be pointed out that the anomalies of Figure 4 shows negative anomalies ($-0.1$ SD) in the period 2001–2004 throughout the study area that corresponds to the event of extreme drought recorded in this period in Quito, the capital city [94].

*4.2. NDVI Analysis—Climatic Variables*

The results of the linear and multilinear Pearson correlations at different NDVI temporal levels with precipitation, temperature, and water availability show only a partial explanation of the behavior of tussock grasses. It was observed that vegetation has positive linear correlations with precipitation, negative with temperature and high R-squared with discharge flows. In addition, the multilinear correlation of NDVI with precipitation and temperature show higher Pearson coefficients than the linear correlations with these individual variables. The results show that the vegetation correlations at large temporal scales (6 and 12 months) have higher average coefficients, except for the linear correlation of NDVI and precipitation at 12 months (Table 7). These results are confirmed by the study by Vega-Jácome [99] on the Peruvian Andes, which shows higher correlations at the same time scales. Zones 1 and 6 showed the highest Pearson coefficients and the highest rates of significance data (Table 8), which are in agreement with the results obtained in wetlands in Peru [100]. On the other hand, it has been observed that NDVI has a better correlation with evaporation [101], the high mountain vegetation in Peru has a greater influence of water availability, temperature, cloudiness, solar radiation, slope angle and elevation [102–104], and the biomass of the Antisana grasslands is inversely related to elevation [91].

Laraque et al. [105] showed the spatial regimes of discharge in Ecuador are related to precipitation patterns. Of the two main slopes of the Andes, the slope towards the Amazon has fewer fluctuations in water flow [106]. In addition, the Pacific slope during the first half of the year shows greater water availability, whereas the Amazon slope has peak flows in the middle of the year. The R-squared between the monthly mean NDVI and monthly mean discharge shows no linear relationships. These relationships are different from the correlations of NDVI and precipitation, which show higher correlation coefficients in the Western Range. Likewise, the coefficients of determination between discharge and NDVI show, that at the time of increasing discharge (April–May, peak rainfall), the level of greenness also increases [105,107]. Thus, water availability is a dominant factor in the greenness of grasslands, as corroborated by the results of the high Andean vegetation [99,100].

Extensive studies have been conducted on different types of vegetation and their relationship with temperature on a global scale. There is a relationship between the root,

stem, and leaf biomass [108]. Other studies on trees located at elevations higher than ~1700 m.a.s.l. showed a positive trend between their diameter and temperature [109]. In the case of Ecuadorian tussock grasses, the relationship with temperature is negative (weak and moderate), whereas greenness decreases as the temperature rises. For both precipitation and water availability, the greatest relationships were observed throughout the Western Range.

*4.3. NDVI Analysis—Global Climate Indices*

Various studies on the Andes have confirmed that climatic variables are directly or indirectly linked to global indices [74,110–119]. The study correlated greenness with various global climate indices, among which the North Atlantic Oscillation (NAO) had the highest Pearson coefficients. This index is related to the trade winds that originate in the north tropical Atlantic Ocean, which previous studies have concluded to be strongly correlated with precipitation in Colombia [112,117]. The results showed an increase in the average Pearson coefficients in the direction of the Pacific Ocean in correspondence with the direction of the trade winds.

On the other hand, all the indices that are based on sea surface temperature of the Pacific Ocean (ENSO, MEI, PDO, El Niño 1 + 2, 3, 3.4 and 4) showed their main incidence in the climatology of the Colombian coasts, Ecuador and Peru [2–4,44,115,120–124]. Of these indices, PDO showed weak correlations with NDVI. Studies have determined that the temperature of the Andes is associated with the positive phase of the P DO [125] and ENSO events [113–116]. In addition, the strength of these ENSO events is influenced by the phases of the PDO [74,111], and coastal (Andean) precipitation is linked to El Niño 1 + 2 (El Niño 3.4) [115,124]. Similarly, the Ecuadorian tussock grasses showed a slight response to ENSO events (Figure 4), corroborating the results observed from NDVI in South America and Colombia [43,44] and refuting those obtained in Ecuador and Peru [110,126]. Subsequently, it was observed that the tussock grasses around snow-capped volcanoes have positive NDVI anomalies that are linked to the warm phase of ENSO (March–April), where a negative balance is produced in the glaciers, which promotes their melting and increases the availability of water; thus, the greenness in the tussock grasses increases (Figure 1) [127].

Finally, the Antarctic low-pressure westerly winds measured by the AAO showed weak correlations with NDVI and a small number of cells with significant data, of which the largest are over the Royal Range, corroborating the results of Leeuwen et al. [32]. In contrast, the MJO is considered as one of the main modulators of climatic variability in the tropics [128]. Studies in the Colombian Andes have determined that Rossby and Kelvin waves are linked to precipitation and temperature [119,123,129]. However, the results showed that NDVI not only shows weak correlations with this index, but also that the Ecuadorian Andes do not correlate significantly with the MJO.

**5. Conclusions**

The study examined the spatial and temporal variability of NDVI in 10 equatorial tussock grassland areas. The grasslands displayed consistent characteristics, with the inter-Andean Valley above 2° S latitude, and the Royal Range being the most productive. Peak productivity coincided with maximum rainfall (MAM). In zone 8 (southern inter-Andean Valley), productivity responded to Amazon moisture influenced by Atlantic Ocean temperature, with a peak during JJA.

The NDVI of the Ecuadorian tussock grasslands showed correlations with precipitation and soil temperature, and multivariate analysis revealed stronger associations. Higher precipitation and water availability corresponded to increased greenness, whereas higher soil temperatures were associated with lower NDVI. Anomalies exhibited a northeasterly increase, with negative anomalies in the Western Range owing to the topographical barrier of the Royal Range. The Royal Range had the strongest positive tendencies, influenced by moisture from the Amazonian region. Positive trends were observed throughout the

study area, and potentially linked to global $CO_2$ fertilization. The linear relationships between NDVI and teleconnection indices were generally weak, with NAO demonstrating the highest ratios. The grasslands showed a slight response to El Niño and La Niña events, warranting further investigation using non-linear models.

**Author Contributions:** Conceptualization, X.Z.-R.; Methodology, J.V.-V. and X.Z.-R.; Validation, J.V.-V.; Formal analysis, J.V.-V., K.U.-Z. and C.B.-E.; Investigation, J.V.-V. and X.Z.-R.; Writing—original draft, J.V.-V. and X.Z.-R.; Writing—review & editing, J.V.-V. and X.Z.-R.; Supervision, X.Z.-R.; Project administration, X.Z.-R. All authors have read and agreed to the published version of the manuscript.

**Funding:** The authors gratefully acknowledge the financial support provided by the Escuela Politécnica Nacional (National Polytechnic School) for the development of the project PIJ 17-05: "Los patrones climáticosglobales y su influencia en la respuesta temporal y espacial de índices espectrales de la vegetación delpáramo en el Ecuador" (Global climate patterns and their influence on temporal and spatial responses ofthe Ecuadorian Páramo vegetation's spectral indices).

**Institutional Review Board Statement:** Not applicable.

**Informed Consent Statement:** Not applicable.

**Data Availability Statement:** The data presented in this study are openly available in the following data repositories in: NDVI—MODIS: https://ladsweb.modaps.eosdis.nasa.gov/ (accessed on 4 March 2019), UCSB-CHIRPS: http://chg.geog.ucsb.edu/data/chirps/ (accessed on 25 Febrary 2019), LST&E-MODIS: https://lpdaac.usgs.gov/products/mod11b3v006/ (accessed on 16 August 2019), AAO: www.cpc.ncep.noaa.gov/products/precip/Cwlink/daily_ao_index/aao/ (accessed on 21 February 2020), MEI: psl.noaa.gov/enso/mei/ (accessed on 21 February 2020), MJO: www.cpc.ncep.noaa.gov/products/precip/CWlink/daily_mjo_index/proj_norm_order.ascii (accessed on 30 January 2020), NAO: www.cpc.ncep.noaa.gov/products/precip/CWlink/pna/nao.shtml (accessed on 23 January 2020), PDO: www.ncdc.noaa.gov/teleconnections/pdo/ (accessed on 24 January 2020), El Niño 1 + 2: psl.noaa.gov/data/correlation/nina1.data (accessed on 22 January 2020), El Niño 3: psl.noaa.gov/data/correlation/nina3.data (accessed on 22 January 2020), El Niño 3: psl.noaa.gov/data/correlation/nina4.data (accessed on 22 January 2020), El Niño 3.4: psl.noaa.gov/data/correlation/nina34.data (accessed on 22 January 2020).

**Conflicts of Interest:** The authors declare no conflict of interest.

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
