# Peer review of "Spatio-Temporal Description of the NDVI (MODIS) of the Ecuadorian Tussock Grasses and Its Link with the Hydrometeorological Variables and Global Climatic Indices"

_sustainability, doi:10.3390/su151511562_

Round 1
Reviewer 1 Report
Dear Authors,
This article focuses on the importance of a specific vegetation type in paramos regions and how to evaluate its change over time.
Generically the text is very clear and I found that the English is generically very correct.
Abstract
The abstract seems consistent with the contents of the article.
Introduction
The literary review is very comprehensive and uses an adequate number of references.
The objectives and research questions of the research are clearly presented.
In the title you mention the study of a specific vegetation community, the Tussock Grasses, however, the objectives and research questions are not specificaly about this vegetation context. Are there other types of vegetation in the region? To my understanding, although the Tussock Grasses areas the dominant element, there are other types of vegetation in the study areas. Please clarify.
Materials and Methods
This section is complete and very understandable.
Figure 1 is rather confusing, not only because of the order of the elements in the collection, non-sequential, but also because one cannot easily locate the sections in the map, because of the overlapping elements.
Some of the contents in Table 4 are still in Spanish.
Results
This study uses a remarkable amount of data.
Results are clearly presented.
Conclusion
In the conclusions many questions are repeated. In my opinion you should consider removing them from this section.
Specific corrections
Line 352 – A reference is made to Figure 4, should be Figure 5
Line 492 – Do you mean: Lower surface temperature.
Kind regards,
Author Response
Dear Authors,
This article focuses on the importance of a specific vegetation type in paramos regions and how to evaluate its change over time.
Generically the text is very clear and I found that the English is generically very correct.
Abstract
The abstract seems consistent with the contents of the article.
We agree. We appreciate your comment.
Introduction
The literary review is very comprehensive and uses an adequate number of references.
The objectives and research questions of the research are clearly presented.
Thanks for your comment.
In the title you mention the study of a specific vegetation community, the Tussock Grasses, however, the objectives and research questions are not specificaly about this vegetation context. Are there other types of vegetation in the region? To my understanding, although the Tussock Grasses areas the dominant element, there are other types of vegetation in the study areas. Please clarify.
We agree with this comment. The research question has been modified as follows: (i) What is the spatial and temporal variability of NDVI at different locations within tussock grasses throughout the equatorial Andes? Please find this change on page 3 lines 138 and 139.
Three main vegetation types: tussock grasslands, cushion plants, and desert-like vegetation are dominant within the páramo ecosystem. We have extensively discussed the dominant vegetation types on page 4 starting on line 172.
Materials and Methods
This section is complete and very understandable.
We appreciate your comment.
Figure 1 is rather confusing, not only because of the order of the elements in the collection, non-sequential, but also because one cannot easily locate the sections in the map, because of the overlapping elements.
We have change Figure 1. Please see new Figure 1 on page 5. The main changes we have made are the following:
- We have changed the name of sequence of panels in Figure 1.
- In new panel c) we have enlarged font sizes of all the 3 páramo sections and we have erased all the unnecessary information.
- In pane d) we show only the tussock grasses extend and in blue dots we are representation the location of water discharge stations.
- Figure 1 description was updated.
Some of the contents in Table 4 are still in Spanish.
The entire content of the table is presented in English. Please see new Table 4 on page 8.
Results
This study uses a remarkable amount of data.
Indeed. We have worked and processed 1242 MODIS images to cover all the tussock grasses extension within the Ecuadorian Andes. We have explained the number of mosaics created on page 10 line 326.
Results are clearly presented.
We appreciate your comment.
Conclusion
In the conclusions many questions are repeated. In my opinion you should consider removing them from this section.
Research questions that were written in the first draft within the conclusions section were erased in the new draft. The conclusion section has been condensed. Please see changes on page 18 starting in line 584.
Specific corrections
Line 352 – A reference is made to Figure 4, should be Figure 5
As we had many figures in the manuscript as one of the reviewers pointed out, we have permanently erased Figure 5. Instead, now we are showing the same information on Table 6 on page 11. The reference has been changed due to the fact that we have deleted that figure. Now we reference the table of the same information. The new reference sentence is on page 11 line 351.
Line 492 – Do you mean: Lower surface temperature.
The sentence has been modified as suggested to avoid. Please see the new sentence on page 16 line 475.
For the overstated reasons I am recommending that this article should be accepted with minor revision.
We appreciate all your comments. Thank you for your time.
Kind regards,

Reviewer 2 Report
This study examined the changes in tussock grass greenness using NDVI data and studied how hydrometeorological variables (precipitation, soil temperature, water availability) and climatic indices (AAO, MEI, MJO, NAO, PDO, El Niño 1+2, 3, 3.4 and 4) influence greenness dynamics in 10 key areas of the paramo ecosystem in the Ecuadorian Andes. The paper is well organized and written. It is important and significant, but some revisions need to be done before the manuscript can be accepted.
Comments:
1. Line 487, there is no title for Table 11.
2. Line 294, the authors should describe DJF, MAM, JJA, SON refers to which months, respectively.
3. The Conclusions section is too lengthy and need to be revised in the general format, for example, with two or three points. Questions in this section can be addressed in the Introduction section.
4. There are too many Figures and Tables in the manuscript with a total of 21.
Author Response
This study examined the changes in tussock grass greenness using NDVI data and studied how hydrometeorological variables (precipitation, soil temperature, water availability) and climatic indices (AAO, MEI, MJO, NAO, PDO, El Niño 1+2, 3, 3.4 and 4) influence greenness dynamics in 10 key areas of the paramo ecosystem in the Ecuadorian Andes. The paper is well organized and written. It is important and significant, but some revisions need to be done before the manuscript can be accepted.
We appreciate all your comments.
Comments:
- Line 487, there is no title for Table 11.
The Table name is Table 10 (continued). Please see on page 16.
- Line 294, the authors should describe DJF, MAM, JJA, SON refers to which months, respectively.
The months referred to on line 294 are described as follows in the revised version of the manuscript: “December-January-Febrary: DJF; March-April-May: MAM; June-July-August: JJA; September-October-November: SON”. Please see changes on page 9 lines 293, 294 and 295.
- The Conclusions section is too lengthy and need to be revised in the general format, for example, with two or three points. Questions in this section can be addressed in the Introduction section.
For the conclusion section we have omitted displaying research questions and we have shortened the text. Please see the new conclusions on page 18 starting on line 584.
- There are too many Figures and Tables in the manuscript with a total of 21.
We agree. We have deleted Figure 5, 6, 8 and 10 on the first draft. Those figures were erased because table 6 contains the same information. Please see Table 6 on page 11.

Reviewer 3 Report
In lines 51 and 52, what does the phrase "compression of the climate on regional scales" refers to?
In the Introduction section, lines 56 and 57, the authors state that there is an information gap related to the interactions between climate and vegetation. However, lines 119 and 120 are written below: "...strong relationships have been found between vegetation indices and climatic variables..." Please clarify this idea.
Figure 1 is confusing in its actual form. Please consider flipping up and down images to get letters in order a)-b)-c)-d)
Table 4 presents lines in Spanish; please translate and consider using the foot page for the URLs.
What does "N" in Equation 1 stand for?
Figures 5, 6, 8, 9, and 10 have a "c)" unnecessary.
Table 11 does not have a title description.
Please do an exhaustive writing review. It is evident that the document was written in Spanish and then translated into English without considering appropriate technical grammar. Avoid the use of the first person of plural (words like "we" and "our"). This action could reduce the amount of information repeated with different sentences about the same ideas.
Author Response
In lines 51 and 52, what does the phrase "compression of the climate on regional scales" refers to?
Thanks for your comment. We have rephrased the sentence for a better understanding. Please see page 2 lines 51 and 52.
In the Introduction section, lines 56 and 57, the authors state that there is an information gap related to the interactions between climate and vegetation. However, lines 119 and 120 are written below: "...strong relationships have been found between vegetation indices and climatic variables..." Please clarify this idea.
The paragraph in lines 56 and 57 alludes to the information gap regarding hydroclimatic variables that significantly influence the high mountain Andean vegetation. On the other hand, the paragraph in lines 119 and 120 pertains to explain that on different ecosystems in different parts of the globe strong relationships have been found between vegetation indices and climatic variables. Therefore, we posit in our study that vegetation dynamics in the paramo ecosystem (tussock grasses) represented by changes in NDVI might also reflect changes in climatic and water availability.
Figure 1 is confusing in its actual form. Please consider flipping up and down images to get letters in order a)-b)-c)-d)
We have changed Figure 1. Please see new Figure 1 on page 5. The main changes we have made are the following:
- We have changed the name of sequence of panels in Figure 1.
- In new panel c) we have enlarged font sizes of all the 3 páramo sections and we have erased all the unnecessary information.
- In pane d) we show only the tussock grasses extend and in blue dots we are representation the location of discharge stations.
- Figure 1 description was updated.
Table 4 presents lines in Spanish; please translate and consider using the foot page for the URLs.
The entire content of the table is presented in English in the revised version of the manuscript. Please see new Table 4 on page 8.
What does "N" in Equation 1 stand for?
Equation 1 has no N. In Equation 2 N is the number of months during the study period. We have added this description. Please see page 9 line 299.
Figures 5, 6, 8, 9, and 10 have a "c)" unnecessary.
Thanks for your comments. Figures 5, 6, 8 and 10 were deleted.
Section "c)" has been removed from new figure as follows:
- Figure 6. Peaks of quarterly maximums of NDVI in the grasslands. For each pixel, the quarter where the maximum NDVI values usually occur has been selected. The year was divided into the following quarters: DFJ, MAM, JJA, SON according to Ecuadorian seasonality. Please see page 13.
Table 11 does not have a title description.
The Table has been named to “Table 10. (continued)”. Please see page 16.
Please do an exhaustive writing review. It is evident that the document was written in Spanish and then translated into English without considering appropriate technical grammar. Avoid the use of the first person of plural (words like "we" and "our"). This action could reduce the amount of information repeated with different sentences about the same ideas.
The writing grammar underwent a thorough review.
